# pMHC Structural Comparisons as a Pivotal Element to Detect and Validate T-Cell Targets for Vaccine Development and Immunotherapy—A New Methodological Proposal

**DOI:** 10.3390/cells8121488

**Published:** 2019-11-22

**Authors:** Priscila Vianna, Marcus F.A. Mendes, Marcelo A. Bragatte, Priscila S. Ferreira, Francisco M. Salzano, Martin H. Bonamino, Gustavo F. Vieira

**Affiliations:** 1Laboratory of Human Teratogenesis and Population Medical Genetics, Department of Genetics, Institute of Biosciences, Federal University of Rio Grande do Sul, Porto Alegre 91.501-970, Brazil; privianna@gmail.com; 2Laboratory of Bioinformatics (NBLI), Department of Genetics, Institute of Biosciences, Federal University of Rio Grande do Sul, Porto Alegre 91.501-970, Brazilmarcelobragatte@gmail.com (M.A.B.); 3Program of Immunology and Tumor Biology, Division of Experimental and Translational Research, Brazilian National Cancer Institute, Rio de Janeiro 20231-050, Brazil; priscila.souzaf@gmail.com (P.S.F.); mbonamino@inca.gov.br (M.H.B.); 4Laboratory of Molecular Evolution, Department of Genetics, Institute of Biosciences, Federal University of Rio Grande do Sul, Porto Alegre 91.501-970, Brazil; francisco.salzano@ufrgs.br; 5Vice Presidency of Research and Biological Collections, Fundação Oswaldo Cruz, Rio de Janeiro 21040-900, Brazil; 6Laboratory of Health Bioinformatics, Post Graduate Program in Health and Human Development, La Salle University, Canoas 91.501-970, Brazil

**Keywords:** viral epitopes, cellular immunology, cancer targets discovery, immunotherapy targets

## Abstract

The search for epitopes that will effectively trigger an immune response remains the “El Dorado” for immunologists. The development of promising immunotherapeutic approaches requires the appropriate targets to elicit a proper immune response. Considering the high degree of HLA/TCR diversity, as well as the heterogeneity of viral and tumor proteins, this number will invariably be higher than ideal to test. It is known that the recognition of a peptide-MHC (pMHC) by the T-cell receptor is performed entirely in a structural fashion, where the atomic interactions of both structures, pMHC and TCR, dictate the fate of the process. However, epitopes with a similar composition of amino acids can produce dissimilar surfaces. Conversely, sequences with no conspicuous similarities can exhibit similar TCR interaction surfaces. In the last decade, our group developed a database and in silico structural methods to extract molecular fingerprints that trigger T-cell immune responses, mainly referring to physicochemical similarities, which could explain the immunogenic differences presented by different pMHC-I complexes. Here, we propose an immunoinformatic approach that considers a structural level of information, combined with an experimental technology that simulates the presentation of epitopes for a T cell, to improve vaccine production and immunotherapy efficacy.

## 1. Background

### 1.1. Structural Immunoinformatica in Viral Infections—The Rational Basis

Viral infections are challenging and are under constant investigation, since the rate of viral mutations is frequently extremely high [1,2,3,4,5,6]. Therefore, immune responses must be fast, but also specific, to avoid viral particle replication and dissemination. Some sets of immune system cells are engaged in the eradication of these pathogens, such as CD8+ cytotoxic T cells, the central players in fighting infected cells [7]. In addition, humoral cell response plays an important role in the elimination of circulating viral particles through neutralizing antibodies, for example. During infection, viruses invade target cells, starting to produce their proteins for the assembly of new virions. In this scenario, a sample of these pathogen proteins is mainly degraded by immunoproteasomes, generating small peptides (usually 8 to 12 amino acids) which are translocated by a transporter associated with antigen processing (TAP 1/2 structure) to the endoplasmic reticulum, emerging on the cell surface associated with MHC class-I molecules. This pathway also includes ubiquitinated cytosolic self-proteins, as well as tumoral ones.

Several bioinformatics tools predict the critical steps of the antigen processing and presenting route, including proteasome cleavage points, TAP translocation aptitude, MHC binding, and even the most probable immunogenic peptides in a putative protein [8,9,10,11]. Nevertheless, the final immunogenicity triggering of an epitope results from the appropriate interaction of the T-cell receptor (TCR) with the pMHC contact surface atom combination [12,13,14,15,16,17]. Thus, a comprehensive understanding of the process demands more than a simple comparison of immunogenic and non-immunogenic peptide sequences, as it usually occurs in the development of predictors [9]. It demands a deeper analysis, from a structural point of view. Evidence for this comes from evidence that very similar peptide sequences can generate dissimilar pMHC surfaces (that come into contact with the TCR), while nonrelated peptide sequences could present almost identical pMHC surfaces (regarding topography and charge distribution, key elements for TCR recognition) [18,19,20]. An understanding of these sequences/structure correspondences will make it possible to infer immunogenic fingerprints, as well as autoimmunity and cross-reactivity trigger identification. In this scenario, a high-scale analysis of peptide:MHC-I (pMHC-I) molecules at a structural level is pivotal to understanding the molecular mechanisms underlying immunologic recognition during infections.

The use protein structure repositories, like the Protein Data Bank (PDB) (https://www.rcsb.org), and the application of large-scale bioinformatics modeling tools could help to elucidate the hallmarks that elicit an adequate cytotoxic immune response. This information can be used to guide tetramer synthesis of putative immunogenic pMHCs that were previously screened by our computational structural analysis, to determine whether they could be effective in the viral clearance. Briefly, our proposed rationale is based on the evaluation of pMHC-I structures of virtually all the peptidome from a prospective virus presented in the context of a specific MHC allele, modelled through our reliable approach, named DockTope tool (http://tools.iedb.org/docktope/) [21]. An alternative method, aiming to reduce the computational cost, is to recover the top scored peptides predicted by different antigen processing pathways tools [22], to be modeled in Docktope. These structures should be compared with pMHC-I structures of previously-described immunogenic targets, which are already contained in our CrossTope Database [23]. This comparison could be performed by hierarchical clustering analysis, for example, as depicted in [18,19], or by a direct electrostatic potential data comparison through MatchTope (tool in development, personal communication). The prospective peptides from viral proteins, which are highly similar to immunogenic ones, are recovered as the most promising targets, and these peptide sequences are used to synthesize the tetramers for posterior testing (workflow summarized in Figure 1). This same approach (in silico structural prediction + tetramer synthesis of promising targets) could be applied, in an alternative rationale, where immunodominant viral epitopes variants can be investigated to observe structural or physicochemical variation and their impact on the cross-reactivity and viral escape [24]. This procedure would be extremely useful for the establishment of immunization strategies in emerging viral diseases and infections caused by viruses presenting high rates of mutations [25], where the synthesized tetramers could be used to validate the generated hypotheses. An interesting example would be the search for targets in the Zika virus, where there is little information about the CD8+ T cell epitopes involved in the elicitation of cellular responses in humans, even five years after the last outbreak. The same pipeline as that described above can be applied to searching for CD4+ helper T cells epitopes. It would be very interesting to investigate whether the fingerprints present in epitopes recognized by cytotoxic T cells are shared by those that trigger T helper responses. Nevertheless, DockTope is unable to model class-II peptides. As our approach considers allele-specific structural patterns, they are easier to find in the Class-I MHC cleft (given its particular structural features). The MHC-II cleft molecule presents the extremities opened and more linear surface topography, allowing the accommodation of a greater range of ligands sizes. It hinders the prediction of the binding core, hampering the extraction of more subtle electrostatic potential footprints, for example. Some methods are improving the benchmarks, and we hope that our pipeline will be used for this important purpose [26,27,28,29]. The knowledge of which viral peptides are involved in a specific infection response and its contribution to the disease progression is essential for an understanding of disease control and further pathogen elimination. The whole proteome of the intended virus can be approached by our proposed pipeline to determine the most promising targets. Tetramers containing these targets can be tested against samples of infected individuals to confirm that they really are the immunogenic epitopes of the infection. With this fast and straightforward screening method, reliable targets can be listed and made available to be applied in a spectrum of immunotherapeutic approaches.

### 1.2. In Immunogenic Tumor Peptides

As we know, in recent decades, cancer has been identified as an important public health problem all over the world. In 2018, in the United States, it is estimated that 1,762,450 new cancer diagnoses and 606,880 cancer deaths occurred, i.e., more than 4600 new cases of cancer were diagnosed each day [30]. In Brazil, the estimated new number of cases were 634,880 in 2018 [31]. This pathological process is mainly caused by a combination of mutations that alter cell functions and growth, generating a malignant neoplasm. With tumor growth, the rate of mutations also increases, which have the potential to generate new peptides (neoepitopes) in tumor cells that can be recognized by the immune system as foreign [32,33,34]. These peptides are tumor-specific, which, once presented by HLA class-I molecules, can trigger specific cellular immunity and generate a specialized repertoire, a crucial element to overcome tumoral heterogeneity [35]. Together with other proteins present only in the affected tissue or organ, like the oncofetal antigens, the mutated class-I peptides are parts of a class named tumor-specific-antigens (TSAs), potential targets to elicit and potentiate immunity in cancer [36]. In this context, cytotoxic T lymphocytes (CTL) play an important role in orchestrating the recognition and elimination of tumor cells [37,38]. As stated above, like viruses, cancer cells have the ability to escape immune surveillance through their high mutational capabilities [39,40]. Nevertheless, some of these variations may have a negative impact on the tumor fitness. In this sense, there are constraints that can be investigated and exploited in immunotherapeutic approaches. Tumor-specific mutations can be recognized as neo-antigens by the T-cell repertoire, since the healthy tissues do not express it. However, these mutations present a unique profile of expression in each patient, making it difficult to use these antigens as generic therapeutic targets [41]. Furthermore, some amino acids or physicochemical changes in the aforementioned cancer epitopes can impact their immunogenicity. Again, the effect of tumor-specific mutations, as in viral epitopes, cannot be accessed just by sequence examination. We must also consider the structural and physicochemical changes in the neoepitope mutations, as well as their impact on T-cell receptors. These different approaches can be achieved by the proposed structural in silico method combined with other computational data analysis tools that facilitate the screening of TSAs and the prediction of neoantigens that could be used in prognostic or therapeutic approaches. Also here, tetramer technology can be applied to give a picture of the real T-cell repertoire in each specific cancer type or even for individual cancers, as well as to select the best immunogenic peptides (the predicted targets that effectively stimulate a T-cell response) to establish a personalized vaccine design to be applied in distinct tumor types. Based on an initial analysis of DNA/RNA sequencing from the individual tumor cells, followed by the inference and modelling of putative immunogenic peptidome and analysis of the structural impact caused by changes in amino acid sequence variants, the synthesis of these peptides (or putative targets) using tetramer technology makes it possible to identify tumor-specific T cells, providing a fingerprint for each tumor type [42].

### 1.3. A Proof of Concept to Test the Putative T-Cell Targets: Tetramers Technology

Tetramer technology was first described in 1996 by John Altman [43], and helped scientists in studies of antigen-specific T-cell responses. This method consists of the synthesis of complexes of peptide-MHC class I (pMHC-I) over a fluorescently-labeled streptavidin to mimic an antigen-presenting cell. These molecules can be customized to select specific T cells, where each complex can be composed of the desired MHC class-I allele (e.g., HLA-A*02:01), synthesized in conjunction with a specific epitope. The principle of tetramer technology is based on the fact that most T cells recognize nonself endogenous peptides derived from a virus, tumor, or other pathogens that are processed and presented by MHC class I on the nucleated cell surfaces [44]. As this recognition is specific for both MHC allele and peptide, the tetramers are able to recover antigen-specific cells. This technology gave rise to a better understanding of CD8+ cytotoxic T-cell phenotypes, dynamics, and functions in a variety of conditions [45,46,47,48]. Here, we propose the use of tetramers in cancer therapy and/or vaccine design, combining the rational prediction of efficient immunogenic peptides associated with the testing of those targets through tetramer assays. This workflow will allow us to recruit and/or identify in patient samples specific and effective T cells against the central targets involved in viral infections and the tumorigenesis of different cancers, which could then be applied in vaccine development and cancer immunotherapies. Also, tetramer technology could be used in combination with humoral immunity approaches, since the peptides presented via MHC-I are derived from the endogenous proteins of virus or tumor cells, while the antibodies generated by humoral response are specific and effective to external proteins.

### 1.4. Target Prediction and Cross-Reactivity

To be feasible, tetramer synthesis should be based on the proper immunogenic target prediction to improve the odds of finding the triggers that elicit an immune response. As previously discussed, these effective T-cell targets could be selected based on reliable immunoinformatics prediction tools [8]. Many studies aiming to predict immunogenic epitopes focus on MHC binding predictions as the most important step in the antigen processing pathway (APP). In fact, to be an immunogenic target, all peptides must be good binders for the correct presentation and interaction with the TCR. Nevertheless, not all good binders are immunogenic targets, considering the omnipresence of self-peptides that are continuously presented on the nucleated cell surfaces, which fortunately, does not elicit an immune response in most cases. The entire immunogenic requirements involve important additional steps in the process that must be performed by peptides, like their correct generation by the proteasome and their translocation by TAP proteins into the endoplasmic reticulum, where they can be attached to the MHC cleft. Nonetheless, the whole process only guarantees that this peptide will be presented, but not necessarily that it will trigger a cytolytic event [9]. Again, the use of predictors that simulate these different steps is important, as it solves part of the immunogenic puzzle. We must understand how T-cell receptors differentiate self and nonself pMHC complexes [9,15,49,50,51]. It seems that the molecular aspects contained in these complexes, such as particularities in topography and charge distribution, are involved in immunogenicity discrimination [18,19,20]. They probably influence the TCR/pMHC interaction and stability that can culminate in the death of the virus-infected or tumoral cell. So, to determine whether an immune response will be triggered by a peptide, an additional level of analysis is required: a structural one, with the inference that pMHC complexes present the infection signals that can be correctly interpreted by the lymphocytes. In this sense, several T-cell epitopes have already been described as being immunogenic in different types of tumors or viruses [22], and can be used as hallmarks of infection/immunogenicity.

If we consider the virtual number of immunogenic antigens in different viruses or tumor cells with which the immune system could be faced, as well as the vast but limited repertoire of immune cells that a specific individual could present, the response should be plastic, adaptable, and effective. One mechanism that can compensate for the numeric limitation of the immune system response is cross-reactivity at the T cell level [52,53]. This cross-reactivity allows a single T lymphocyte to recognize distinct peptides derived from the same protein, from different proteins of the same virus, or even peptides from heterologous viruses or heterogeneous sources. After priming with a peptide, i.e., related or analogous ones, or even peptides with low levels of identity with the first contact peptide, can activate the same T-cell clone. Although cross-reactivity has become a well-recognized phenomenon in recent decades, we cannot say the same about the structural basis governing heterologous T cell recognition. We performed studies which demonstrated molecular variations in the TCR contact surfaces of pMHC complexes, and their impact on the stimulation of cytotoxic responses [18,19]. These in silico approaches explained previously in in vitro and in vivo assays were not elucidated by single comparisons of epitopes amino acid sequences [54]. Another work of our group unveiled, exclusively by pMHC structural analysis and comparison, an epitope from the Epstein-Barr virus (CLGGLLTMV_LMP2-426_) that primes T cells of control individuals, giving a false-positive test result for Hepatitis C infection, through cross-reactivity with HCV wild type epitope CINGVCWTV_NS3-1073_ [55]. This is an example in which it could be hard to find the hidden cross-reactive target by a sequence alignment search alone, for example. The implication of structural elements in the pMHCs complexes and its importance for the stimulation of cross-reactivity was extensively demonstrated in Antunes et al. [56]. Structural investigation of the included tumoral targets seems to be fundamental to maximize the efficacy of immune responses and to avoid autoimmune reactions with self-antigens resembling the chosen targets.

Therefore, the combined use of APP (antigen processing pathway) prediction tools with structural analyses of the most promising targets from viral genomes or tumoral proteins can fill the commonly-missing holes of immunogenic epitope prediction. Also, as we have developed a method to model pMHC complexes in a fast and reliable way [21], we can evaluate the impact of mutations in the immune response outcome, or verify if apparently-dissimilar peptides present analogous TCR interaction surfaces, which could stimulate responses against the wild type and cognate peptide. This investigation would be virtually impossible or would be very expensive under regular experimental conditions (like large-scale X-ray diffraction of pMHC complexes). The use of tetramers harboring potential immunogenic peptides selected by bioinformatics could be used in immunotherapy and vaccine design, allowing scientists to gain better knowledge of the elements that orchestrate immunogenicity

### 1.5. The Proposed Method Applied to Viral Vaccine Development

Based on our previous evidence, we propose an innovative pipeline to be applied in vaccine development and immunotherapies against virtually any pathogen or specific cancer. It starts with the sequencing of a viral genome from a new emergent disease, for example (Figure 1A). Considering the molecular aspects of the pMHC complexes that dictate TCR stimulation and consequent cytotoxic triggering, our main idea consists of the full comparison of any possible epitope derived from the intended pathogen genome against a panel of immunogenic targets, from a structural T-cell receptor perspective. That is, virtually the whole viral peptidome/ligandome will be anchored in an MHC structure from a prevalent allele through a molecular modeling approach developed by our group, thus generating the TCR interacting surfaces for each these putative epitopes (Figure 1B,D,E). These surfaces are then compared by clusterization methods with pMHC complexes of the aforementioned targets for the same MHC allele (such targets are considered infection fingerprints) (Figure 1F,G). Peptide sequences from the investigated genome that present similarities with the immunogenic ones are assumed to be the most promising CD8+ T cell targets, and are selected for tetramer synthesis and testing (Figure 1H,I). The proposal for the use of tetramers as our proof of concept is reinforced by the fact that these molecules simulate the molecular presentation context of these putative targets, mimicking in vivo recognition (through the interaction between lymphocytes and the presenting cells). Additionally, in the tetramer synthesis step, we can define the specific allele for which the immunogenic prediction was made. So, we can then argue that the chosen pMHC complexes are able to interact with the cytotoxic T lymphocyte from the sample of infected individuals (Figure 1J). A differential in this kind of approach is the fast and reliable identification of T-cell epitopes that can be envisaged in two hypothetical situations: (i) in an emergent viral disease, where nothing of the causative agent in terms of T-cell targets is known, and where the sequencing of a single sample can be the starting point to infer the viral proteome. This genome/proteome can then serve as an input to our pipeline. In such a situation, an initial structural pMHC comparison could be performed against previously-described targets from viruses belonging to the same family of the newly-described virus (i.e., Zika virus proteome against already-described Flaviviridae epitopes); and (ii) in a disease caused by a highly-mutable pathogen. In such infections, it is very hard to infer the impact of a mutation in an immunodominant epitope by comparing mutated and wild-type sequences alone. Our method can be adapted to verify whether the amino acid substitution alters the TCR interaction surface properties, which usually disrupts recognition.

### 1.6. On Immunotherapy

Immunotherapy is an effective tool for different types of cancer, showing positive results such as selectivity, long-lasting effects, diminished side effects, and improved overall survival and tolerance [57]. The idea of using the immune system to fight cancer was developed in 1992, when William Coley treated sarcoma patients with toxins from bacteria and observed a 10% rate of cancer remission [58,59,60]. This approach resulted in the activation of tumor-circulating leukocytes, the production of inflammation mediators, antibodies, and consequently, tumor elimination. However, despite these great results, immunotherapy is far from being the perfect solution in the fight against cancer [61]. Scientists verified that the immune system becomes tolerant to cancer cells due to their similarities with healthy cells, as well as the expression of regulatory molecules used to evade immune system surveillance. Thus, tumor–immune tolerance must be “broken to combat cancer” [62].

The fight of the immune system during the whole tumorigenic process resembles, to some extent, the combat performed in a viral infection [63]. With subtle differences, the players, as well as the involved molecules, are similar. Therefore, it is possible to extend our hypothesis, proposed for reliable viral vaccine development, to be applied in cancer immunotherapeutic approaches. However, the antigens that are being processed and presented by tumor cells are composed of self-proteins. Thus, the focus should be the aforementioned search for epitopes in tumor-specific proteins and neoantigens in the tissue of the tumor of interest, considering individual variation [32,64,65,66]. Following this rationale, families of proteins that have aberrant expression in different tumor types, also known as cancer/testis antigens (CTA), like Melanoma Antigen Genes (*MAGEs*), Sarcoma Antigen Genes (*SAGEs*), Carcinoembryonic Antigens (CEAs), and New York Esophageal Squamous Cell Carcinoma 1 (NY-ESO-1), have been revealing their importance in the prognoses of different types of cancer, and can be used as promising targets in immunotherapy [64,67,68]. Some studies have shown that these proteins may also be important for different aspects of tumor biology, such as for the development of metastasis [69,70,71]. Mendonça et al. [72], when investigating the roles of MAGE-A10 in squamous cell carcinoma cell lines, demonstrated that this protein is indeed involved in the epithelial to mesenchymal transition (EMT). This relation should be explained by the interference with the elements of the cytoskeleton. Thus, it is believed that the MAGE family may aid in the process of tumor metastasis. Beyond their role in tumor biology, CTA proteins are usually expressed during the embryonic phase, and are aberrantly re-expressed in tumors [73,74,75]. These proteins can represent whole new sets of tumor-associated antigens (TAA), since the entire protein is potentially suitable for immunogenic peptide generation. Thus, the pivotal point here is to discover which are the hotspots of T-cell immunogenicity in these antigens. Indeed, these rich TAA collections generated from CTA can be exploited for tumor vaccination strategies, as have been largely reported for some antigens such as NY-ESO1 [76], MAGEs [77], or PRAME [78]. Proteins such as CTAs are expected to have poor central tolerance, and their expression can be detected by T cells in a way that is similar to the aforementioned recognition for microorganism antigens.

Other antigens detected in tumors can derive from mutations that occur in the carcinogenesis process. These mutations can be divided into mutations that contribute to tumor growth, dubbed “drivers”, and mutations considered benign or neutral (known as passenger mutations). Mutations in important genes which are expressed in several tumors, such as *TP53*, *BRCA1*, and *PTEN*, among others, are considered driver mutations [79]. Such driver mutations represent the minority of mutational events, while passenger mutations correspond to 97% of the mutations in cancer [66,80,81]. Although passenger mutations poorly impact tumor progression individually, the high number of such mutations and the cumulative effects of their large numbers mean that they also contribute to tumor progression [82,83]. A study by McFarland et al. [84] refutes the passenger paradigms as neutral mutation, and suggests that these mutations, are, in fact, detrimental to metastatic progression, preventing or retarding the growth of micrometastasis. Other studies support the results found by McFarland et al., showing that passenger mutations impair tumor development [85,86,87,88] and reduce cell proliferation [89,90]. Following this rational, it has been demonstrated that a greater number of passenger mutations may be related to better results in cancer therapies [86,91,92,93]. These putative targets generated by punctual mutagenic processes can be better probed and accessed through combined new sequencing methods with structural methodologies, as proposed here, by their subtle nature.

Tumor-associated antigens were once expressed in the embryonic stage of development, before lymphocyte selection in the thymic microenvironment, thus being considered nonself antigens from an immunological point of view. On the other hand, tumor neoantigens are generated as a byproduct of the somatic mutations that occur concomitantly with tumor development, being highly immunogenic due to a lack of central tolerance [65]. Tumors that have a high neoantigen load, such as melanoma, have greater chances of response to immunotherapy [94,95], with T lymphocytes reacting to MHC class-I [96,97,98] and -II restricted neoantigens [99]. Nonetheless, only a small fraction of non-synonymous mutations in the genes expressed in tumors will lead to the formation of neoantigens triggering reactivity of CD4+ and/or CD8+ T cells. As a general rule, a load of 10 mutations per megabase (Mb) of DNA seems to be enough to lead to the frequent formation of neoantigens that will be recognized by the immune system, being considered a high mutation load [95]. Another important factor to consider when it comes to immunotherapy with immune checkpoint blockade, that ultimately relies on antigen specific T cell recognition, is that patients who have mutations leading to neoantigens being present in the stem of the clonal tree have a better chance of responding to therapy because a larger number of tumor cells will have the same neoantigen, ultimately leading to tumor regression [100]. A higher level of complexity is added in determing the tumor epitope, in relation to viral infection, when we consider tumor heterogeneity [95,101]. A viral similarity with cancers comprises cases of chronic infections, where different strains could coexist, as in HIV and HCV infections, explaining the difficulty of finding effective vaccines [102]. These problems can be solved by new sequencing technologies, like RNAseq, which can provide information on the tumor heterogeneity of a specific individual [103]. This individualized tumor information could be then used to identify a customized T-cell targetome.

## 2. Discussion and Conclusions

Our approach has strong potential to address the different sources of variability provided by viral and tumor proteins, analyzing which among them possess the best hallmarks of infection or is least vulnerable to the lack of immunogenicity, as a consequence of mutations, from a structural and lymphocytic perspective. The best targets could then be applied to develop a diagnostic test for early cancer detection, for example, where a tetramers kit harboring immunogenic epitopes commonly presented in that cancer will serve to show that a specific individual is mounting an immune response against that antigen, a signal that a tumorigenic process is underway. This procedure is adequate because the antigen processing pathway is constantly presenting a sample of the intracellular proteome, which could include mutated or differentially-expressed proteins. Tetramer technology has also been used in immunotherapy. One possibility is the recovering and expansion of specific effector CD8+ T cell populations, which are responsive to immunogenic epitopes for that tumor, in a customized way of fighting. Fehlings et al. have demonstrated that tetramers loaded with known immunogenic epitopes in conjunction with mass cytometry assays are able to identify specific and reactive intratumoral neoantigen-CD8+ T cells pools [45]. The idea is to use some epitopes or a cocktail of epitopes to select and expand CD8+ T cell lymphocytes that can fight tumor cells for the tumor proteome of that specific individual, efficiently covering the heterogeneity and preventing genetically-altered cell populations from escaping and giving rise to new tumors. As an example of T cell stimulation and tumor recognition, tumor-infiltrating T lymphocytes (TILs) were used along with interleukin-2 and lymphodepleting chemotherapy to treat several metastatic melanoma patients, leading to objective clinical responses in 51% of patients, including four who had complete tumor regression [104,105]. While such an approach has repeatedly shown impressive results in high mutation burden tumors, such as melanomas, achieving similar results has been challenging in less mutated tumors. A recent report described a similar approach, where T lymphocytes that respond to specific peptides of mutated proteins from a patient with metastatic breast cancer were expanded in vitro. These cells were thus re-infused along with PD-1 (programmed cell death 1) blockade, leading to tumor eradication for more than 22 months, and indicating that, although labor intensive, this therapy modality can potentially be applied and have a role as therapy in less immunogenic tumors [106].

Nowadays, immunotherapy focuses on the use of self-cells that are expanded and enhanced in their antitumor capabilities. These cells are potential antitumor killers and are obtained from the same patient that will be treated. However, it is hard to predict in which patient and/or cancer type immunotherapy could be effective. In case of failure, it is essential to perform detailed and retrospective studies, to identify where and why this strategy failed. Tetramer technology can help in identifying potential antitumor cells that could be used in immunotherapy. After the prediction, selection, and synthesis of tumor antigens, these can be coupled with tetramer structures and incubated with blood samples from a patient with cancer. After this step, the tumor-specific cells linked to tetramers could be harvested and expanded in vitro. In this way, potent antitumor cells will be highly enriched to be used to fight the tumor. Alternatively, the T-cell receptor of the recovered cells can be sequenced and used to generate Chimeric Antigen Receptor T cells directed against immunogenic peptides that are specifically present in that individual sample or, generically, against the shared tumor antigens present in a specific tumor type. The goal of tetramer technology ranges from validating the discovery of new, tumor-specific antigens to time-saving in fighting cancer. However, predicting tumor targets is not a simple task. The peptide target by the T lymphocyte on the adoptive cell therapy must be expressed only in the tumor, because otherwise, it could cause unwanted reactions. Initially cancer-testis antigens (CTA) seemed to be interesting targets, because this family of genes is expressed in several types of cancer. However, that enthusiasm was attenuated by the low level of protein expression on these antigens (only 10% of cancers maintain sufficient protein expression to be targets suitable for antitumor T cells), and by the unanticipated low expression of these antigens in healthy tissues [94]. This on-target/off-tumor recognition can lead to potential deleterious side effects. Cameron et al., using TCRs targeting MAGE-A3, generated a cross response to MAGE family proteins and to Titin, a muscle peptide, causing fatal immune reactions [107]. A different approach for generating such antitumor responses is selecting tumor-specific T cells based on tetramer selection. One of the limitations of such an approach is that although the use of tetramers for tumor target identification is a viable strategy, the frequency of T cells which are specific for tumor-associated immunodominant antigens in the circulation is low, rendering this approach technically challenging. When a tumor is discovered, all clinical approaches should be applied in a short period of time.

Computational tools able to predict potential tumor or viral antigens in conjunction with an efficient method by which to select responsive T cells are essential for eliciting anti-cancer/anti-viral effects. In this regard, our approach presents a new way to detect reliable targets, highlighting the importance of finding molecular fingerprints of immunogenicity that can be used in immunotherapy and vaccine development in combination with current immunoinformatics predictors.

## Figures and Tables

**Figure 1 cells-08-01488-f001:**
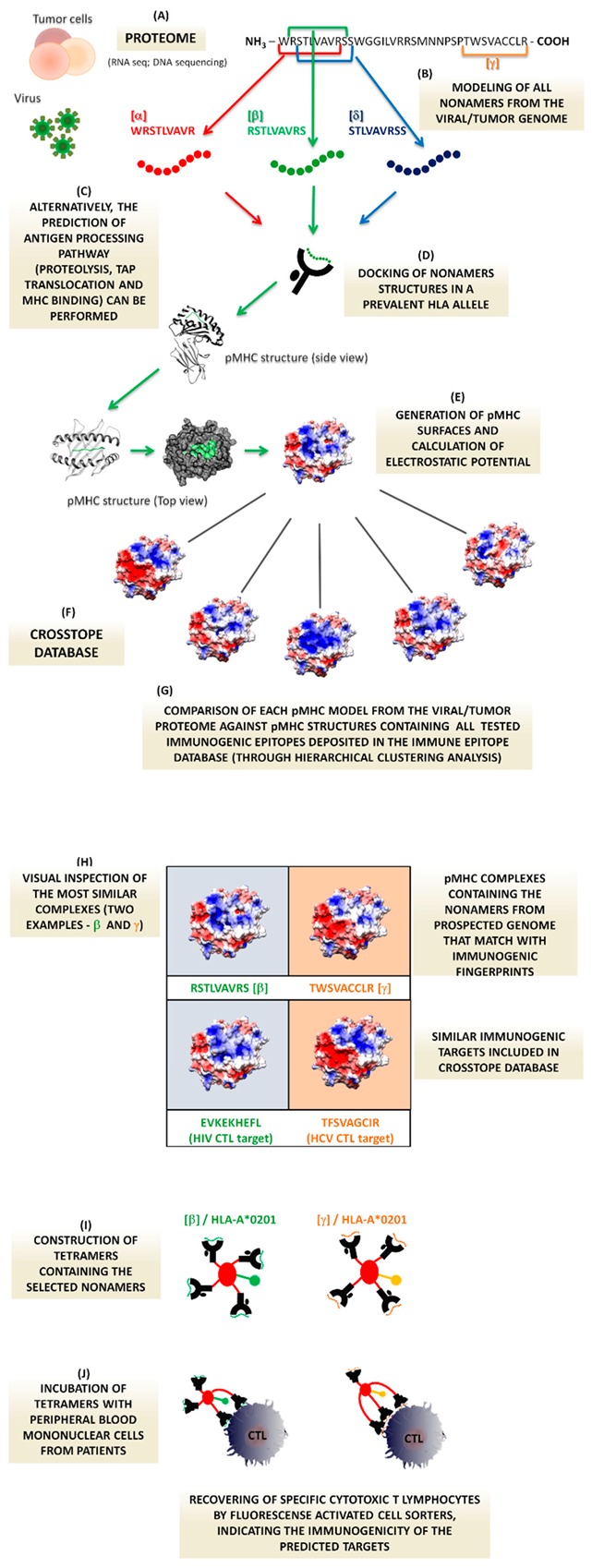
(**A**) From the sequencing of both DNA or RNA from tumor cells or viruses, it is possible to infer the proteome from these samples. (**B**) These protein sequences can be decomposed into nonamers through the sliding window (In the figure, α, β, δ, and the last nonamer (γ) are represented), which make it possible to model the complete cellular peptidome in a prevalent MHC allele of interest (e.g., HLA-A*02:01). (**C**) Alternatively, the complete protein sequence may be subjected to predictors of the antigen processing pathway. (**D**) Docktope will be used to generate the pMHC complexes obtained by overlapping the entire peptidome or by the selection of the best-ranked epitopes, docking the peptides in their MHC allele-specific conformation. (**E**) The electrostatic surface will then be calculated for all the three-dimensional structures of the modeled pMHCs (negative charges in red, positive charges in blue, and neutral in white). (**F**) A second set of data will consist of all the electrostatic surfaces contained in the Crosstope Structural Databank, which contains pMHC structures displaying the sequences of immunogenic epitopes described in the literature for prevalent human alleles. (**G**) The pMHC charge distributions of the investigated nonamers will be compared with fingerprint electrostatic surfaces, looking for shared patterns, which will indicate the most promising regions of the proteome of interest that induce an immune response. (**H**) Examples of two putative targets are displayed and can be verified by simple visual inspection. Above, the electrostatic surfaces of the nonamers β and γ are shown, and below, the similar immunogenic targets of β (EVKEKHEFL/HIV epitope) and γ (TFSVAGCIR/HCV epitope) are indicated. The determination of these similar targets between proteomes of interest and immunological databases is performed by hierarchical clustering methods which make it possible to compare hundreds of targets. (**I**) The sequences of candidate nonamers presenting similarities with the immunogenic targets will be used for the synthesis of tetramers (putative sequence and the respective MHC allele for which it has been modeled). (**J**) For the proof of the concept, the synthesized tetramers will be incubated with blood samples from patients infected with the studied virus or with the tumor of interest. This approach would allow us to find targets that are usually recognized in cancer or viral infections in a customized and fast way. These targets can then be used in diagnostic or immunotherapeutic approaches.

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
