# Peer review of "pMHC Structural Comparisons as a Pivotal Element to Detect and Validate T-Cell Targets for Vaccine Development and Immunotherapy—A New Methodological Proposal"

_cells, 2019, doi:10.3390/cells8121488_

Round 1
Reviewer 1 Report
In this review by Vianna and co-authors, an overview of the most recent improvements in methodologies to identify and validate tumoral or viral epitopes is described in detail. This review is timely and well described, given the strong interest in uncovering new antigens that can be used as novel immunotherapeutics in the field of infectious disease and cancer. A flowchart in Figure 1 is easy to follow, touches on important aspects of peptide discovery and validation methods. Rationale for describing their new methodology is sound, and conclusions/discussion include various clinical applications that can be translated from these discovery methodologies. In general, this review is very well written, up-to-date and the authors clearly have expertise in this area. A lingering question to this reviewer is how peptide/MHC technologies differ for those that bind MHC-1 (which is the focus of this review) to those that bind MHC-II and are recognized by CD4 T cells? This is a minor point, which can be addressed with a small paragraph in the concluding statements.
Author Response
We thank you for the comments on the review and the recognition of our work. I agree that a paragraph discussing the application of MHC-II epitopes can improve the manuscript quality. The only issue in pMHC-II modeling is the difficulty in define the core region to model since there is a variable number of amino acids that a class II peptide can be composed of. This more permissive range of peptide length is conferred by an MHC cleft opened in its extremities in the class II molecules. In this sense, a short discussion was included to cover this important area of the immune response, performed by T helper cells. The paragraph included was: "The same pipeline above described can be applied to searching for CD4+ helper T cells epitopes. It would be very interesting to investigate if the fingerprints present in epitopes recognized by cytotoxic T cells are shared by those that trigger T helper responses. Nevertheless, DockTope is unable to model class II peptides. As our approach considers allele-specific structural patterns, they are easier to be found in the class-I MHC cleft (given its particular structural features). The MHC-II cleft molecule presents the extremities opened and more linear surface topography, allowing the accommodation of a greater range of ligands size. It difficulties the prediction of the binding core, hampering the extraction of more subtle electrostatic potential footprints, for example. Some methods are improving these benchmarks and we hope in the future our pipeline could be used for this important purpose." in the "Structural Immunoinformatics in Viral Infections - the Rational Basis" section.
Reviewer 2 Report
The authors here claim that the virtual modeling of the structure of the surface of MHC-peptide complexes that is presented to the TCR allows a more reliable prediction of T-cell epitopes and their crossreactivity than conventional T-cell epitope prediction tools. In contrast to a previous paper with similar conclusions from the authors (Antunes et al, 2017, Front Immunol), this manuscript does not include data to support their claims. The paper contains a mixture of a basic introduction to cancer immunotherapy and tetramertechnology together with speculations of the potential applications of their structure-based epitope prediction tool. These applications do not differ from other T cell epitope prediction tools.
The mistakes found in English grammar, make some parts difficult to understand.
Less basic immunology and general speculations but data that support the claims made would improve the manuscript.
Author Response
The comments raised by the referee are very intriguing and deserve some discussion. While there is some resemblance in the manuscript from Antunes et al. (2017) with this work, the focus and the objective of both works are distinct. In that work, the objective was to discuss the cross-reactive events in murine and human sets of peptides, by a structural point of view and its potential consequences in immunotherapeutic approaches. There is no mention of the use of large-scale structural comparison of a prospected viral, bacterial or tumoral proteome with previously immunogenic targets as an alternative method to search for putative epitopes. The hypothesis proposed in this work is a synthesis of elements discussed in other articles published by our group. In these works, we presented many sets of previously published cross-reactive epitopes and the effect of physicochemical similarities in the differential stimulation of immunogenicity. It evidences that there are structural fingerprints of immunogenicity and these elements are present in widely recognized immunogenic peptides (deposited in CrossTope and recovered in Immune Epitope Database). The presence of these similar elements in prospected proteomes is indicative of potential immunogenic targets. So, we use elements and observations extracted in our previous studies discussing cross-reactivity to propose an original pipeline to prospect T cell targets from a structural perspective. So, in this sense, the proposed hypothesis is embased on experimental data already presented and reproduced in our previous articles, which are listed below.
Structural Allele-Specific Patterns Adopted by Epitopes in the MHC-I Cleft and Reconstruction of MHC:peptide Complexes to Cross-Reactivity Assessment. https://doi.org/10.1371/journal.pone.0010353
Structural in silico analysis of cross-genotype-reactivity among naturally occurring HCV NS3-1073-variants in the context of HLA-A*02:01 allele.https://doi.org/10.1016/j.molimm.2011.03.019
Improved structural method for T-cell cross-reactivity prediction. doi: 10.1016/j.molimm.2015.06.017.
CrossTope: a curate repository of 3D structures of immunogenic peptide: MHC complexes. doi: 10.1093/database/bat002.
Interpreting T-Cell Cross-reactivity through Structure: Implications for TCR-Based Cancer. Immunotherapyhttps://doi.org/10.3389/fimmu.2017.01210
Reviewer 3 Report
This a good review article presenting latest information on the subject. The authors are requested to address the following issues:
The authors only mention software developed by them. They should also mention the ones developed by others.
Abstract: What is El Dorado? Please use a scientific or standard English word. Many people may be unfamiliar with it.
Line 45: assembling=assembly
Line 58 and elsewhere: Evidences; never used as a plural word
Line 88: few=little
Line 135: add , after …..class I molecules
Line 156: Stimulate, and not stimulates
Line 177: posteriorly is not an appropriate word in the context
Line 201: culminate in, and not with
Line 244: consists of, and not on
Line 289: Explain abbreviations
Line 293: The sentence requires appropriate punctuation.
Line 297: embrionary=embryonic
Author Response
We would like to thank you for the valuable observations of the referee. All the suggestions were implemented or changed. References and mentions on other tools were included. Concerning tools developed by other groups, all the cited approaches to perform predictions on the antigen-presenting pathway and to predict T cell targets by sequence analysis were developed by researchers around the world. Nevertheless, we include other tools to model pMHC complexes trying to fill the gap suggested by the referee. The term "El Dorado" refers to a mythic city full of gold lost in some part of the Amazon forest, searched by many conquerors. We intended to compare the searching for reliable targets with this very hard quest made along many centuries. If it is not a good term, we can change it by a more familiar expression. We wanted to provoke in the reader the curiosity to know a new term/history. I am sure that our manuscript was improved with your suggestions.
Round 2
Reviewer 2 Report
The authors have developed a tool to model and compare the contact surfaces presented by MHC-I/peptide complexes to the TCR. They claim that by the analysis of structural similarities to known MHC-I/peptide structures, potential new immunogenic peptides and cross-reacivity between peptides with dissimilar sequences that produce similar contact surfaces can be predicted. The tool developed for this purpose is DockTope. The general idea is not new and the basics already presented in D.A. Antunes et al, Front Immunol., 2017, but is now described in more detail and breadth in the submitted manuscript. The hypothesis that cross-reactivity may be better predicted when structural data are included is indeed obvious and should be further tested. Thus this paper could fuel research in this direction. However, as the current T cell epitope prediction programs have become more sensitive and specific, it remains unclear if the authors believe that the predictions programs will become more sensitive or specific when adding the DockTope tool. The authors should describe in more detail the limitations of the current prediction tools and discuss if and how the structural analysis will potentially improve specificity or even sensitivity.
Author Response
First of all, we understand that the work seems to present some overlapping concepts with our previous articles, but it is important to note that the purposes and the expected results are distinct.
DockTope tool is our tool to model pMHC customized complexes, in a fast and reliable way, not requiring computational skills from the users. It was published in Scientific Reports and now is hosted on the Immune Epitope Database. It does not have the epitope clustering function based on structural similarities.
The work from D.A. Antunes et al., 2017, published in Frontiers in Immunology, discussed the importance of structural similarities in pMHC complexes to explain previously published cross-reactive networks and the importance in investigating that in immunotherapeutic approaches, to avoid autoimmune reactions. There is no mention in the whole text about T cell epitope prospection using this rationale. Nevertheless, we agree that this may not have been clear in the text and we changed these sentences by "The implication of structural elements in the pMHCs complexes and its importance on the stimulation of cross-reactivity was extensively demonstrated in Antunes et al., 2017 [56]. The structural investigation of the included tumoral targets seems to be fundamental to maximize the efficacy of immune responses and to avoid autoimmune reactions with self-antigens resembling the chosen targets."
In the current article, we propose that if there are structural similarities shared by described epitopes, they can be the hallmarks of immunogenicity. So, investigating a candidate viral or tumoral proteomes, searching for these fingerprints is an innovative way to prospect T cell targets. This pipeline is not presented in any of our previous works. We know the importance of current prediction programs. Our approach has no intention to substitute them but to add a new level of investigation in T cell prospection. In this regarding, we included a final paragraph "Computational tools able to predict potential tumor or viral antigens in conjunction with an efficient way to select responsive T cells are essential for eliciting anti-cancer/anti-viral effects in a patient. In this regarding, our approach brings a new way to prospect reliable targets, highlighting the importance of finding molecular fingerprints of immunogenicity that can be used in immunotherapy and vaccine development in combination with current immunoinformatics predictors." The new level refers to the importance of the structure in giving us the fine specificity in particular scientific problems. We can point out the work of Zhang et al., 2015 (DOI: 10.1128/JVI.00539-15), for example. In this work, the discovering that the EBV LMP2-426 (CLGGLLTMV) epitope was the target that elicited cross-reactive response against the WT HCV NS3-1073 (CINGVCWTV) that gave false-positive in HCV-negative individuals was only possible when we observed structural similarities in pMHC surfaces shared between these T cell epitopes. How we could do this prospection in a hypothesis-free situation where there was no idea of which virus/bacteria were the targets for the subsequent false-positive priming?
We included the following sentence in this section of the manuscript: "Another work of our group unveiled, exclusively by pMHC structural analysis and comparison, an epitope from the Epstein-Barr virus (CLGGLLTMVLMP2-426) that primes T cells of control individuals, giving a false-positive test result for Hepatitis C infection, through cross-reactivity with HCV wild type epitope CINGVCWTVNS3-1073. This is an example where could be hard to find the hidden cross-reactive target only by sequence alignment searches, for example."
We should consider that these cross-reactive epitopes do not present conspicuous sequence similarity. In this sense, it is hard to find the hidden cross-reactive target only by sequence alignment search, for example. It is true, even considering good benchmarks for current T cell epitopes prediction tools. Such a task is almost unpractical when we do not know the putative cross-reactive pathogen.
It is difficult to state that our approach will raise the sensitivity and specificity of the cytotoxicity prediction. We believe that the number of tested targets can be reduced with their respective reliability improved, when selected in combination with other prediction programs. The answer could come with research groups testing our hypothesis in their future works. We hope we have been clarified the raised questions.